# 3D molecular generative framework for interaction-guided drug design

Wonho Zhung [1], Hyeongwoo Kim[1] & Woo Youn Kim [1,2,3] ✉

Deep generative modeling has a strong potential to accelerate drug design. However, existing generative models often face challenges in generalization due to limited data, leading to less innovative designs with often unfavorable interactions for unseen target proteins. To address these issues, we propose an interaction-aware 3D molecular generative framework that enables interaction-guided drug design inside target binding pockets. By leveraging universal patterns of protein-ligand interactions as prior knowledge, our model can achieve high generalizability with limited experimental data. Its performance has been comprehensively assessed by analyzing generated ligands for unseen targets in terms of binding pose stability, affinity, geometric patterns, diversity, and novelty. Moreover, the effective design of potential mutant-selective inhibitors demonstrates the applicability of our approach to structure-based drug design.

Incorporating adequate prior knowledge is critical for developing generalizable deep learning models in data-deficient scientific problems[1–3]. The domain-specific prior knowledge of a particular task can help a model featurize generalizable patterns across its training data, leading to noteworthy successes[4–7]. For instance, AlphaFold2[8] utilized the co-evolutionary information to narrow down the extensive conformational space of protein folding on a macroscopic scale and the residue pair representation to reduce the structural complexity on a microscopic scale. The development of a generalizable model for designing novel hit compounds, a primary goal of computer-aided drug design, would be such a task where prior knowledge is necessary[9,10].

The advent of deep generative modeling is changing the paradigm of drug design. Generative models trained with activity data for a specific target protein can design new molecules with strong binding affinity to the protein[11–14]. However, their performance is hampered by the lack of activity data, causing limitations due to their low generalization ability. First of all, generated molecules likely consist of core structures learned in training, making them less innovative[14–16]. This prevents the possibility of identifying promising hits with novel core structures. Second, the designed molecules often interact unfavorably with the target though they are structurally similar to the training molecules with high activity[17–19] because a small change in a molecular

structure often causes a large drop in activity due to the complex interaction patterns between a protein and ligand molecules[20]. Thus, the low generalization ability may result in molecules with low binding stability and affinity. These problems become more severe for newly discovered proteins where little data is available.

To avoid the dependency on limited activity data, recent generative models utilize the 3D contexts of a binding pocket, enabling pocket structure-based ligand design with no reliance on activity data[21–28]. Ragoza et al. represented the electron density of a ligand as voxels and trained their model to reconstruct the voxelized density from the given pocket structure, pioneering a 3D molecular generative model in structure-based drug design[21]. Luo et al. designed ligands by sequentially adding ligand atoms directly inside a pocket, achieving better molecular properties[22]. Zhang et al. attempted to enhance the generalization ability of the 3D generative models by featurizing the local geometric patterns involved in protein–ligand interactions[26], leading to a significantly improved performance in a benchmark study to design potent molecules for unseen targets.

A well-generalized model should comprehend the universal nature of protein–ligand interactions, including hydrogen bonds, salt bridges, hydrophobic interactions, and $\pi$–$\pi$ stackings, which are essential for strong binding stability and affinity regardless of the protein and ligand pair. Proper exploitation of these interaction types

[1]Department of Chemistry, KAIST, 291 Daehak-ro, Yuseong-gu, Daejeon 34141, Republic of Korea. [2]AI Institute, KAIST, 291 Daehak-ro, Yuseong-gu, Daejeon 34141, Republic of Korea. [3]HITS Inc., 124 Teheran-ro, Gangnam-gu, Seoul 06234, Republic of Korea. ✉e-mail: wooyoun@kaist.ac.kr

as prior knowledge can help generalize structure-based drug design models[29–33]. In this context, we believe that local geometric patterns repeatedly observed in various protein–ligand binding structures, used in the previous work[26], imply information about the interaction types, leading to more generalizable models. However, these patterns would not be sufficient to fully exploit the universal features of the interaction types for structure-based drug design, as limited complex structure data cannot cover all possible geometric patterns made by the combination of targets and ligands.

Here, we propose an interaction-aware 3D molecular generative framework that leverages the universal nature of protein–ligand interactions to guide structure-based drug design. While a target pocket can form different combinations of protein–ligand interaction types depending on the binding ligand and its binding pose, we aim to inversely design a ligand that fulfills a specific combination of interactions using a 3D conditional generative model, named DeepICL, which can be applied to any kind of protein. We use local interaction conditions in a subpocket where ligands should be bound, instead of using whole interaction information, to prevent undesirable bias to specific pockets or ligand structures.

To demonstrate the ability of our framework for generalizable structure-based drug design, instead of using a typical benchmark consisting of $10^5$ to $10^7$ computer-generated protein–ligand binding structures[21–26], we use only about $10^4$ ground-truth crystal structures curated from the PDBbind database[34] since a well-generalized model can successfully extract appropriate features even from small-sized data. We assess our model by analyzing various aspects of generated ligands for unseen targets—binding stability, affinity, geometric patterns, diversity, and novelty. Finally, we apply our model to tackle practical problems where specific interaction sites play a crucial role, demonstrating the applicability of our approach to structure-based drug design.

## Results

### Interaction-aware 3D molecular generative framework

Our framework consists of two main stages—(1) interaction-aware condition setting and (2) interaction-aware 3D molecular generation—as illustrated in Fig. 1. Here, we provide a general overview of each stage. More details of each stage can be found in the Method section.

The first stage of the framework aims to set an interaction condition, **I**, by investigating protein atoms of a given binding site, **P**. We used four types of protein–ligand interactions—hydrogen bonds, salt bridges, hydrophobic interactions, and π–π stackings. We only considered the four most dominant interaction types in the protein data bank (PDB)[35], since we used the PDBbind 2020 dataset[34] for model training, which originated from the PDB[36].

Recently, Zhang et al.[31] built a conditional RNN-based molecular generative model that used interaction fingerprints (IFPs) to incorporate protein–ligand interaction information in generating ligands in SMILES. Likewise, we develop a protein atom-wise interaction-aware conditioning strategy. We define an interaction condition as a set of protein atoms' additional interaction type one-hot vectors which indicates whether the atom can be involved in a particular interaction and its role in the interaction. Protein atoms are categorized into one of seven classes—anion, cation, hydrogen bond donor and acceptor, aromatic, hydrophobic, and non-interacting atoms. In contrast to representing entire interaction information as a single interaction fingerprint, our strategy aims to establish interaction conditions locally. As illustrated in Fig. 1b, only neighboring pocket atoms are considered in each step of atom addition; thus, a specific interaction condition of these pocket atoms is utilized.

In this work, we mostly determined pocket atoms' interaction classes in two strategies, as described in Fig. 1a. During the generation phase, information on how a receptor interacts with a ligand is not always available. Thus, we predefined criteria for interaction classes so that we can designate the interaction condition on each protein atom by analyzing them. Since we do not use any reference ligands for condition setting, we call this condition a reference-free interaction condition. For instance, we render SMARTS patterns[37] to determine hydrogen bond acceptors and donors. Detailed criteria for aromatic atoms, hydrophobic atoms, cations, and anions are described in Supplementary Table 3. Meanwhile, there are ground-truth structures of protein–ligand complexes, **C**, during the training phase, so we extract interaction conditions from those reference structures. We used the protein–ligand interaction profiler (PLIP)[38], which is software that

### (a) Stage 1: Interaction-aware condition setting

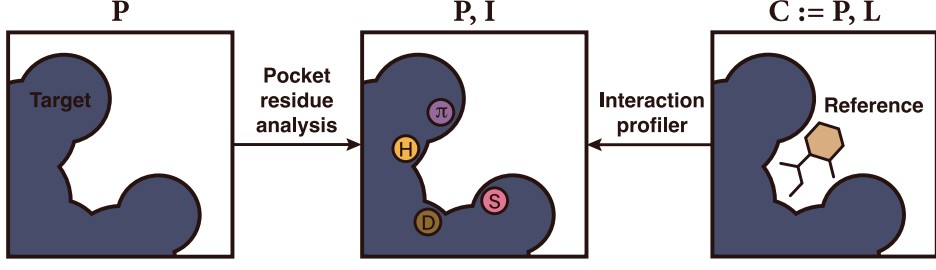

### (b) Stage 2: Interaction-aware 3D molecular generation

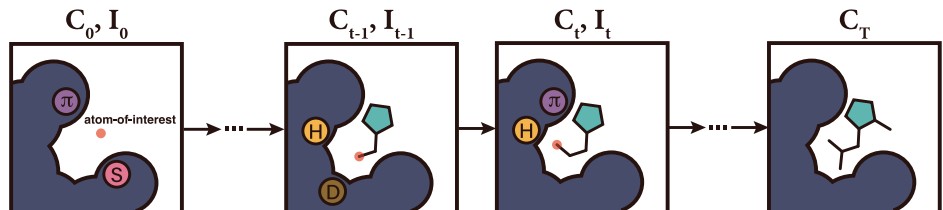

**Fig. 1 | A conceptualized illustration of our proposed interaction-aware 3D ligand generative framework. a** The first stage profiles a protein pocket to designate an interaction condition on each protein atom. **b** In the second stage, DeepICL sequentially adds ligand atoms inside a protein pocket based on the predetermined interaction condition. Letters inside circles indicate interaction types as follows: hydrogen bonds (**H**), hydrophobic interactions (**D**), salt bridges (**S**), and π–π stackings (**π**).

identifies non-covalent interactions between a protein and a ligand by analyzing their binding structure. Besides these two strategies, from the practical perspective, one can manually designate a desired interaction condition from one's insight based on the knowledge of the target system.

In the second stage, atoms in a ligand are sequentially generated based on the 3D context of a pocket and the interaction condition from the first stage. For this purpose, we devised a deep generative model named DeepICL (**Deep I**nteraction-aware **C**onditional **L**igand generative model) for carrying out the inverse design of ligands. As illustrated in Fig. 1b, ligand atoms are added one by one, and the atom of interest changes in each step. This atom of interest indicates a position where the next atom is added. Thus, we only considered the surrounding environment of the atom-of-interest, $C_t$, so that the local interaction condition, $I_t$, is newly defined to feed into the current generation step $t$. We demonstrate our framework on two molecular generation tasks—ligand elaboration and de novo ligand design—inside a target protein. The former task aims to refine a known ligand to improve its potency. The latter aims to design a ligand from scratch, providing diverse molecules that can fit in a binding pocket. Both tasks are crucial in the structure-based drug design but challenging due to the vast number of drug-like molecules, known to be over $10^{23}$ [39], and distinct binding sites present in each protein. For the ligand elaboration task, the binding pose of a ligand core structure is given and used as an initial state. In the de novo ligand design task, one can manually select a point inside a pocket, which serves as a starting point. The detailed architecture of DeepICL is provided in the Method section.

### Effect of interaction-aware conditioned ligand design

We first demonstrate the effect of interaction-aware conditioning on designing specific interaction patterns. In drug design processes, it is crucial to construct specific protein–ligand interactions, as they can be directly related to potency and selectivity. Supposing binding sites where ligands can readily interact are known, a generative framework should be capable of designing a ligand that can favorably interact with these sites. To establish a reasonable guess of those interaction hot spots, we utilized interaction patterns of the reference protein–ligand complexes.

Among the test complexes described in the data section, we chose complexes that possess diverse protein–ligand interactions to demonstrate the effect of interaction-aware conditioning. We selected three proteins, which were bone morphogenetic protein 1 (BMP1), fibroblast growth factor-1 (FGF1), and dihydrofolate reductase (DHFR). From the reference ligands of their original complexes, we extracted initial core structures, which were azabicyclo[2.2.1]heptane, 2-(oxan-3-yloxy)oxane, and benzene, respectively. Core structures were determined based on our visual inspection, removing chains and functional groups to leave the minimal structures composed of single or double rings as shown in Fig. 2a and Supplementary Fig. 1. Figure 2 illustrates three examples of ligand elaboration, where their given interaction conditions are visualized as patches in Fig. 2a. In Fig. 2b, the designed ligands with the highest interaction similarity were depicted, along with the original ligands. Here, the interaction similarity estimates the similarity between the interaction patterns of a generated ligand and the original one. We introduce the precise definition of an interaction similarity in the Method section.

Figure 2c, d depicts interaction maps for the original and designed ligands with the highest interaction similarity, respectively. For the first case, which is the left column of Fig. 2, our model successfully designs hydrogen bonds, $\pi-\pi$ stacking, and salt bridges as the given interaction condition with different motifs. Notably, Fig. 2d-1 shows that the model can generate the thiophene ring instead of the original benzene ring to construct a $\pi-\pi$ stacking with TYR68. It implies that the model learned the characteristics of aromatic motifs that are required to form a $\pi-\pi$ stacking. Although the model added aliphatic carbons near the

hydrophobic PHE157, the distance was slightly larger than the threshold to be profiled as a hydrophobic interaction. For the other two cases, DeepICL also successfully designed ligands that exhibit highly similar interaction patterns with the original ones while generating motifs distinct from the original ligands. More comprehensive discussions of each case are provided in Supplementary section 6.

We further demonstrate the effect of our interaction-aware conditioning strategy in reproducing interaction patterns that were given as conditions for elaborating each ligand core structure. As an ablation setting, we masked interaction conditions to exclude information about the reference interaction patterns in the inferencing process. We fed our model a zero vector with the same size as the original interaction condition as a masked interaction condition. We compared the distribution of interaction similarities between two ligand sets that were generated with either the reference or masked condition in Fig. 2e. It clearly shows a substantial difference between the two distributions in every case, where the ligands elaborated with the respective reference condition achieve much higher interaction similarities. Thus, we justified that our framework is highly controllable, providing ligands with desired interaction patterns by elaborating on the existing ligand. Five more examples of interaction-aware conditioned ligand elaboration are given in Supplementary Fig. 1.

### Demonstrating the generalizability of our framework

In the following sections, we demonstrated the generalizability of our interaction-aware generative framework in structure-based drug design. For the baseline comparison, we devised a model that was trained only on binding structures, without any explicit information about protein–ligand interactions, or to be more specific, without using additional interaction condition vectors for protein atoms. This baseline model might learn some information related to non-covalent interactions based on the atom occurrences but is less likely to be generalized on the typical patterns of non-covalent interactions due to the limited number of protein–ligand pairs in the training set. Therefore, the baseline model inevitably relies on the statistical distribution of protein–ligand binding geometries in determining the type and position of a newly added atom. Here, we named sets of the generated ligands from the interaction-conditioned model and the baseline model with and without interaction information, respectively.

### Binding pose stability analysis

We first carried out short (10 ns) molecular dynamics (MD) simulations to assess the binding stabilities of elaborated ligands. If a ligand forms unfavorable interactions with a target, its binding pose will fluctuate largely in a short time period[40,41]. From a trajectory of ligand poses during the MD simulation, their root-mean-square deviations (RMSDs) are calculated as values representing the binding stability of the ligands. Details of running MD simulations and RMSD calculations are included in Supplementary section 5.

While using the same test pockets and ligand core structures as those in the previous section, we performed a ligand elaboration task but with reference-free interaction conditions. We first filtered novel ligands to ensure that the numbers of their heavy atoms are the same as that of the reference ligand. We note that the greater number of heavy atoms likely induces higher binding affinities. Thus, comparing ligands with the same atom numbers was necessary for fairness. This left less than 50 ligands, and then we randomly sampled 10 ligands for MD simulations. We plotted the averaged RMSD values of the 10 designed ligands along with that of the original ligands during the simulations of each case in Fig. 3a. Notably, the ligands elaborated with the interaction information showed RMSD values as low as the original ligands. This is strong evidence of model generalization; that is, the model was capable of generating ligands that stably bind to unseen targets as effectively as the reference ligands do. Although our model was not trained on the ligand MD trajectory data that explicitly informs

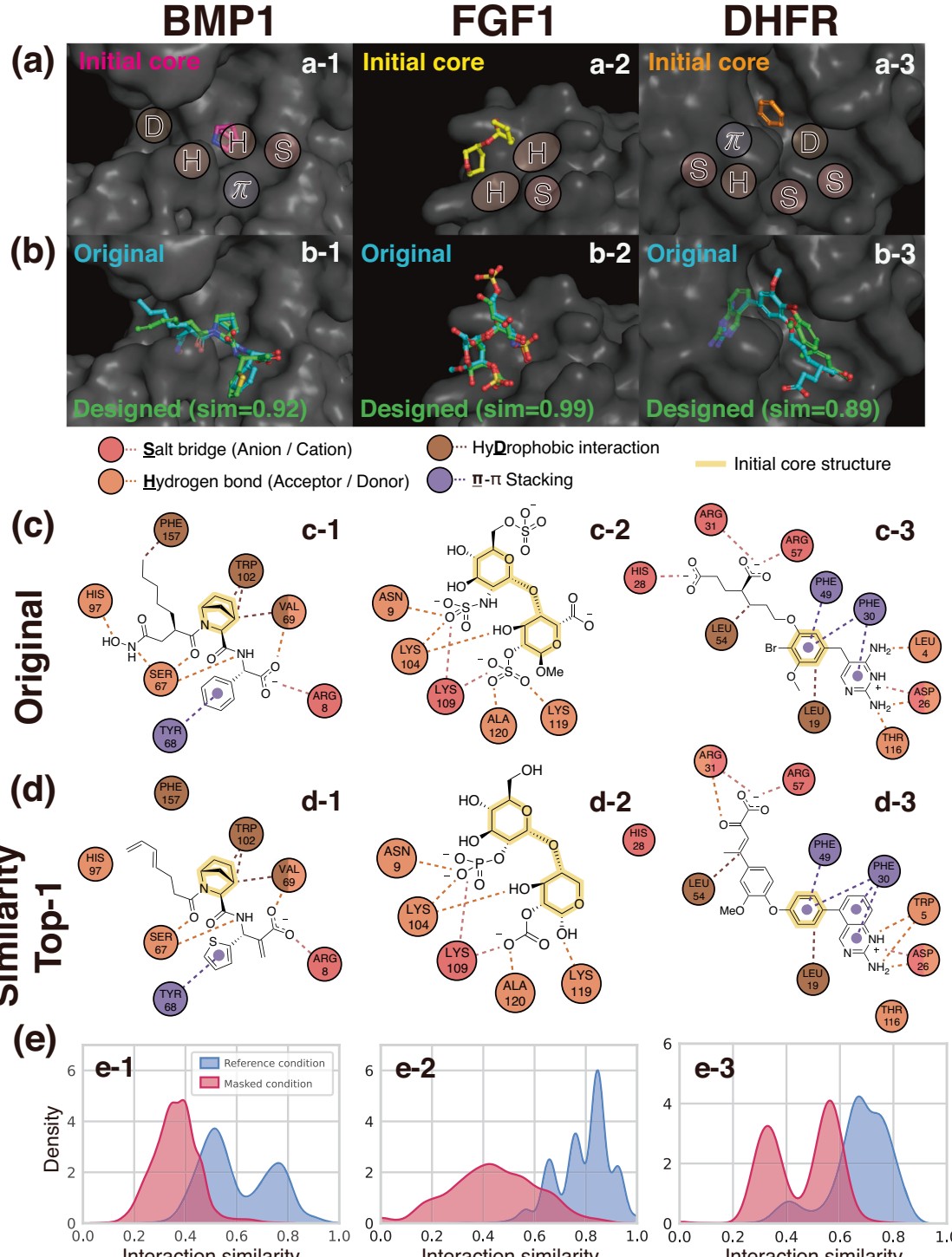

**Fig. 2 | Examples of interaction-aware conditioned ligand elaboration. a** Initial core structures and binding pocket surfaces marked with given interaction conditions. Although an interaction condition is defined at an atom level, we illustrate conditions as patches for a better visual representation. **b** The original and designed ligands of the highest interaction similarities with the respective similarity value. **c, d** The 2D diagrams of profiled interactions between the pocket and the original ligands or designed ligands. The circles indicate amino acid residues, and the dashed lines indicate the interactions. Different colors are used to distinguish interaction types, where circles with multiple colors correspond to the residues involved in more than one type of interaction. The core structures used as an initial structure are highlighted in each ligand. **e** The distributions of interaction similarities of ligands generated with the reference and masked condition. Source data are provided as a Source Data file. Left: bone morphogenic protein 1 (BMP1, PDB ID: 6bto), middle: fibroblast growth factor 1 (FGF1, PDB ID: 3ud9), right: dihydrofolate reductase (DHFR, PDB ID: 1dis).

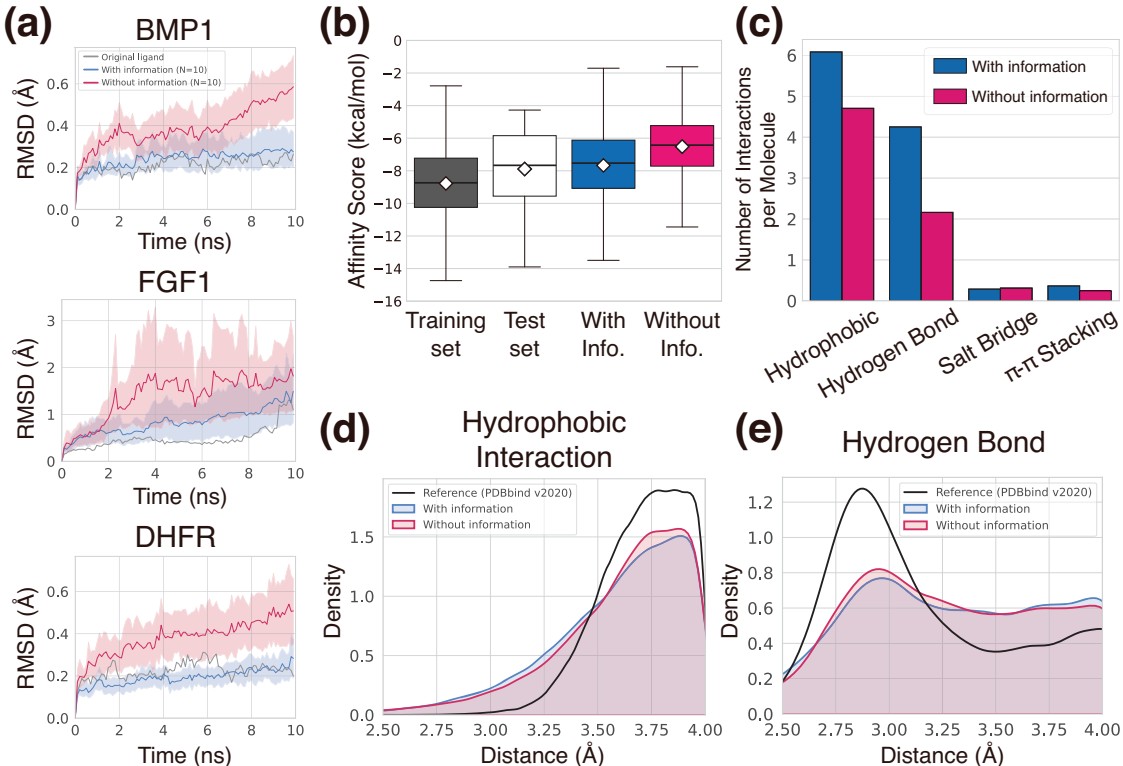

**Fig. 3 | Demonstration of the generalizability of our generative framework.**
**a** Plots of ligand RMSDs during short MD simulations to assess the binding pose stability of designed ligands in three pockets from the test set—BMP1, FGF1, and DHFR. The blue and red curves depict the averaged RMSDs of ten sampled ligands of each generated set with 95% confidence intervals. Gray curves show ligand RMSDs of the original ligands. **b** The binding affinity scores of each set are presented as box plots (center line at the median, upper bound at 75th percentile, lower bound at 25th percentile) with whiskers at the minimum and maximum values. The average scores are also shown as diamonds. 100 ligands were generated with and without interaction information for each of the 100 test pockets, resulting

in a total of 10,000 ligands. Their binding affinity scores were depicted in the blue and red boxes, respectively. The binding affinity scores of the ground-truth complexes composing the training and test sets are also analyzed, and depicted as the black and white boxes, respectively. Note that the training and test complexes are carefully separated, thus the training and test ligands are from distinct protein targets. **c** The bar plot of the number of interactions per molecule for each interaction type from the generated complexes in (**b**). **d, e** Kernel density estimation plots of hydrophobic interaction and hydrogen bonding distances, respectively. The distances from the generated complexes in (**b**) were measured by using the PLIP software. Source data are provided as a Source Data file.

about the binding stabilities, it successfully established stable interactions. Moreover, ligands elaborated with the model without interaction information in all three cases showed higher RMSDs, implying the lack of favorable interactions needed to stabilize their binding poses. The case of BMP1 and DHFR showed clear differences between the two sets, where the ligands elaborated without interaction information showed substantially larger deviations from their initial binding poses. Meanwhile, FGF1 exhibited relatively small differences between the two sets, both showing acceptable stabilities compared to the original ligands. This result suggests that even the baseline model can generate stable ligands to some extent through learning protein−ligand interactions in the training phase.

**Binding affinity analysis of de novo designed ligands**
In addition to measuring how stably the generated ligand molecules are bound to the target, we further analyzed the binding affinities of generated ligands to investigate how strongly the ligands interact with the target protein. We performed a de novo ligand design task instead of the elaboration task done in the binding pose stability analysis. 100 ligands were generated for each of 100 test proteins, resulting in 10,000 ligands. Then, their binding affinities were evaluated with SMINA[42]. The statistics of binding affinity scores for the ligands are illustrated in Fig. 3b. The ligands generated from the baseline model obtained an average value of -6.52 kcal/mol, which was higher than that of the ligands generated with interaction information, −7.67 kcal/mol. Since the lower score indicates stronger binding, the result implies that

the incorporation of interaction information was beneficial to designing molecules with stronger binding affinities. We emphasize that our strategy contributes to improving the generalizability of the generative model by achieving fairly high binding affinities without training on experimental affinity data. The average value of the binding affinities from the training set was -8.78 kcal/mol. Though the average value of the generated ligands with interaction information was higher than that, it is speculated that this was due to the difference in the pocket environments between the training and test sets, because the binding affinity is a function of both protein and ligand structures. Thus, we compared the binding affinity scores of the reference ligands in the training and test sets and found that the protein−ligand complexes in the test set also exhibit higher binding affinity scores than those of the training set. The ligands generated with interaction information and the test set showed similar score distributions, rationalizing that the generated ligands steadily reproduced the distribution of the binding affinities of experimentally verified reference ligands, supporting the above speculation.

Additionally, we analyzed the number of each type of protein−ligand interaction to interpret the role of interaction information in achieving high binding affinities. Since there is no specific value for the desired number of interactions, we compared the generated complexes with and without interaction information for the same test pockets. Using the PLIP software[38], we identified and counted the interactions in the generated complexes. We normalized the counts with the number of molecules to obtain the number of

interactions per molecule, as illustrated in Fig. 3c. In the case of the baseline model, the numbers of hydrophobic interactions and hydrogen bonds were much less than those of the model trained with interaction information and comparable for the salt bridges and $\pi$–$\pi$ stackings. Especially, the number of hydrogen bonds per molecule generated without interaction information was only about half of that with information, where the difference was exceptionally larger than other types of interaction. For example, creating a hydrophobic interaction is relatively straightforward as it involves attaching a non-polar carbon atom in an appropriate location. On the other hand, a hydrogen bond is a directional interaction between a donor and an acceptor, making it more challenging to generate such an interaction type when relying solely on the distribution of a limited number of structural data without prior knowledge on protein–ligand interactions.

In contrast to the baseline model, our framework could generate more hydrogen bonding atom pairs by sampling the next atom in a partially deterministic manner as guided by the prior knowledge of protein–ligand interaction given as a condition. Supplementary Table 4 provides empirical evidence showing that, in a situation where generating a hydrogen bond is favored, the model with interaction information adds nitrogen, oxygen, and fluorine with a higher ratio than that of the baseline model. We attribute the higher binding affinities of ligands generated with interaction information to the higher success rate in hydrogen bonding formation. Although the number of hydrogen bonds might not be directly correlated with binding affinities, its contribution is crucial.

## Geometric analysis of generated interactions

Here, we elucidate how well the modeled distribution reflects the characteristic geometric patterns of protein–ligand interactions, which can be observed in crystal structures of protein–ligand complexes. This can serve as empirical evidence of the model's successful featurization of generalizable patterns from the structure data. However, most recent deep generative modeling approaches for structure-based drug design focused on designed ligands' intramolecular geometry alone, neglecting the analysis of intermolecular geometry. We, thus, demonstrate the geometry of protein–ligand interactions within the sampled complexes. For the generated ligands in the binding affinity analysis, we measured the distances of each non-covalent interaction type without any further structural optimization.

In Fig. 3d, e, we illustrated the geometric distributions of hydrophobic interactions and hydrogen bonds, which were predominant in Fig. 3c. Figure 3d shows a density distribution of hydrophobic interaction distances, the most common type in the PDB[35]. Our DeepICL effectively captured the observed trend of density decaying as the distance decreased. As the distance of hydrophobic interaction is defined between two hydrophobic carbons, the plot shows that the model avoids spatial hindrance while adding a carbon atom. The distances are mostly populated at around 3.8 Å, much longer than hydrogen bonds, in accordance with the observed tendency. Next, Fig. 3e shows a density distribution of hydrogen bonding distances. It is known that heavy atoms involved in a hydrogen bond are separated at a median distance of around 3.0 Å[35]. The distribution from the generated data also shows a peak near 3.0 Å, which is consistent with the tendency. In both types, the distribution was unchanged regardless of the use of the interaction information. The baseline model, trained without explicit interaction information, could also capture the spatial distribution of those interaction types. This implies that the knowledge about chemical interactions was beneficial for the model to know when to generate the right interaction type for a particular binding point, leading to the increased rate of favorable interaction formation rather than forming a more plausible geometric pattern. We provide additional information related to the other two interaction types—salt bridge and $\pi$–$\pi$ stacking—in Supplementary Fig. 3.

## Chemical diversity and novelty of designed ligands

Achieving high chemical diversity and novelty is another essential goal in structure-based drug design, which can be assessed at the level of the core structures or, in other words, scaffolds[43]. Although the overall structure of a molecule is new, if it shares the same scaffold with existing drugs, it may be considered less patentable and, therefore, less likely to be accepted for drug development. Thus, we evaluated the diversity and novelty of Bemis-Murcko scaffolds[44] extracted from generated ligands for unseen targets.

We first evaluated the chemical diversity in terms of the uniqueness of scaffolds among the 10,000 generated ligands with the settings described in the Method section. Out of the 9930 valid molecules, duplicates were removed to yield 5669 unique scaffolds or a scaffold diversity of 57.1%. For comparison, we also assessed the scaffold diversity of the training data. It possesses 5783 unique scaffolds out of 10,752, resulting in a scaffold diversity of 53.8%. Notably, our framework achieved slightly greater diversity than the training data despite using the specific interaction conditions extracted from the references. Further analysis of the frequencies of non-unique scaffolds is provided in Supplementary Fig. 4a.

Then, we evaluated the structural novelty of the designed ligands. In comparison with the training data, 5467 scaffolds are novel among 5669 unique scaffolds, achieving a novelty of 96.4%. This implies that our framework can provide novel structures rather than repeating the core structures learned from the training data. We further compared them with 1,568,892 bioactive compounds in the ChEMBL database[45], of which molecular weights are under 500. 4951 generated ligands possess scaffolds that are not present in the ChEMBL database, indicating that about half of the generated ligands comprise novel scaffolds. These results can be explained by our model's feature that generates atoms considering their local surrounding environment instead of seeing the whole binding region. A few examples of the novel-generated scaffolds are provided in Supplementary Fig. 4b. On the other hand, the high novelty might lead to synthesizability concerns, so we evaluated the synthetic accessibility of the generated ligands by using the SAscore[46] as shown in Supplementary Fig. 5. The SAscores of the generated ligands exhibit a very similar distribution to those of the training set and the known bioactive molecules introduced in the work of Ertl et al.[46]; the average SAscore of the generated molecules was 3.18 which is close to the peak of the graph for bioactive molecules. Based on this result, we can conclude that most molecules designed by DeepICL have structural complexities comparable to those of typical bioactive molecules.

## Site-specific interaction conditioning for selectively controlled ligand design

One of the key advantages of our framework is the capability to establish an interaction condition based on one's prior knowledge, which enables designing a ligand with specific functionality. Here, we chose an important practical problem where forming selective interactions at specific locations is crucial; designing a ligand that can selectively bind to a mutant Epidermal Growth Factor Receptor (EGFR) while sparing the wild-type EGFR. We refer to the Method section for the experimental details of the specific interaction site-conditioned ligand generation process. The binding affinity scores of designed ligands for both the wild-type and the mutated EGFR are illustrated in Fig. 4a with a population density. Points above the solid diagonal line score lower on the mutated pocket than on the wild-type. As a lower score indicates a stronger binding, this tendency clearly shows the mainstream of generated ligands can bind stronger to the mutated EGFR. Since a reduction in energy by 1.36 kcal/mol theoretically corresponds to a 10-fold decrease in inhibitory concentration, we set a difference of 2.72 kcal/mol or a 100-fold difference in inhibitory concentration as the criterion for identifying

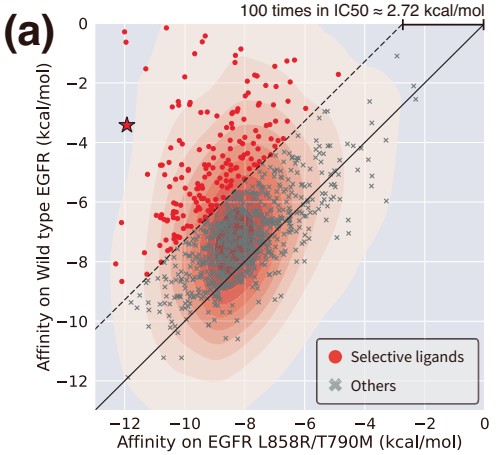

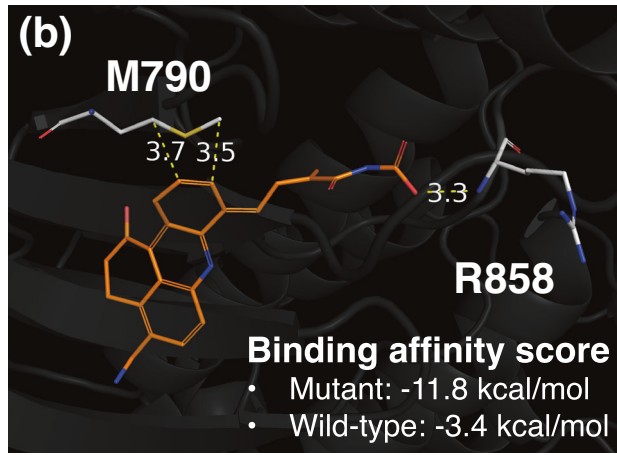

**Fig. 4 | Selectively controlled ligand design via site-specific interaction conditioning. a** The scatter plot illustrating the binding affinity scores of designed ligands toward the wild-type EGFR and the double-mutated EGFR, with their population density. Red points show 2.72 kcal/mol lower binding affinity (corresponding to theoretically 100-fold less inhibitory concentration) for the mutated EGFR. Source data are provided as a Source Data file. **b** An example of a well-designed ligand that is expected to be selective toward the double-mutated EGFR (depicted as a star in (**a**)). Non-covalent interactions and their distances (unit: Å) with mutated residues are shown in yellow dashed lines.

ligands to have selectivity. As a result, we obtained 233 selective ligands, corresponding to red points above the dashed line in Fig. 4a. We selected a well-designed ligand with a visual inspection among the selective ligands and visualized it in Fig. 4b. The ligand forms hydrophobic interactions with a side chain of MET790 while forming a hydrogen bond with a backbone of ARG858. Thus, we could successfully identify a promising molecule that interacts with mutated residues utilizing site-specific conditioning without any training on additional data. We also demonstrated that the site-specific interaction conditioning scheme could design potential hinge-binding ligands of Rho-associated protein kinase 1 (ROCK1) in Supplementary Fig. 6 with the details.

## Discussion

We proposed an interaction-aware 3D molecular generative framework that incorporates prior knowledge of protein–ligand interactions to generalize structure-based drug design. In contrast to previous models that solely rely on binding structural information, we focused on featurizing generalizable patterns of the four types of protein-ligand interactions—hydrophobic interaction, hydrogen bond, salt bridge, and $\pi$–$\pi$ stacking, since they are universal regardless of protein–ligand pairs. We demonstrated the generalizability of our method by comprehensively analyzing the designed ligands for unseen targets in various aspects and confirmed that our framework could establish favorable interactions at a high rate in a controlled manner by virtue of the conditional generative framework. By demonstrating that leveraging protein–ligand interaction types can generalize structure-based drug design, this study suggests that adopting appropriate prior knowledge can improve the generalizability of deep generative modeling in a variety of scientific domains with limited data availability.

## Methods

### Training and test data

In this work, we only used experimental crystal structures from the 2020 version of the PDBbind general set[34] whose binding structures were identified with X-ray crystallography. We split the crystal structure data considering the target sequence similarity so that none of the data pairs between the train and the test has a similarity larger than 0.6, which is calculated and clustered by the CD-HIT software[47]. As a result of data processing, we used 11,284 structures for training our model and 2109 structures for validation. We filtered

out the rest of the data to leave 109 test complexes that satisfy the following three conditions: (1) ligand's Tanimoto similarity is less than 0.6 with all the ligands in the training set, (2) every data corresponds to distinct protein families, and (3) the number of protein heavy-atoms is less than 3000. For convenience, we randomly chose 100 test complexes from them, whose PDB IDs are provided in Supplementary section 12.

### Model overview of DeepICL

The goal of our framework is to model the probability distribution of ligands conditioned on a target protein and the interaction patterns. We represent a ligand and a protein as a set of atoms, $\mathbf{L} := \{L_i\}$ and $\mathbf{P} := \{P_j\}$, respectively. Each ligand atom, $L_i$, is defined as a tuple of an atom type, $\mathbf{X}_i^l \in \mathbb{R}^{F^l}$, and an atom position, $\mathbf{r}_i^l \in \mathbb{R}^3$. Similarly, each protein atom, $P_j$, is represented by an atom type, $\mathbf{X}_j^p \in \mathbb{R}^{F^p}$, and its position, $\mathbf{r}_j^p \in \mathbb{R}^3$. Note that $F^l$ and $F^p$ denote the dimension of atom features for a ligand and a protein, $i$ and $j$ correspond to ligand and protein atom indices, respectively. Details of atom features are summarized in Supplementary Table 1. The main objective is to model a conditional probability distribution, $p(\mathbf{L}|\mathbf{P}, \mathbf{I})$, where $\mathbf{I}$ indicates the interaction condition vector obtained from the first stage of the framework. We factorize the conditional distribution in an autoregressive manner similar to cG-SchNet[48], where the probability of the upcoming ligand atom depends on the existing atoms. By defining a protein–ligand complex at a time step $t$ as $C_t := (\{L_i\}_{i=1}^t, \{P_j\})$, we can formulate the factorization as follows:

$$
\begin{aligned}
p(\mathbf{L}|\mathbf{P},\mathbf{I}) &= \prod_{t=1}^{T} \left[ p(L_t | \{L_i\}_{i=1}^{t-1}, \{P_j\}, \mathbf{I}) \right] \cdot p(\text{stop}|\mathbf{L},\mathbf{P},\mathbf{I}) \\
&= \prod_{t=1}^{T} \left[ p(L_t | C_{t-1}, \mathbf{I}) \right] \cdot p(\text{stop}|\mathbf{L},\mathbf{P},\mathbf{I}),
\end{aligned}
\tag{1}
$$

where $T$ is the number of ligand atoms. $p(\text{stop}|\mathbf{L}, \mathbf{P}, \mathbf{I})$ is a probability of termination, which determines when to stop the generation. We further factorize the conditional probability of a ligand atom at a time step $t$ as:

$$
p(L_t|C_{t-1},\mathbf{I}) = p(\mathbf{X}_t|C_{t-1},\mathbf{I}) \cdot p(\mathbf{r}_t|\mathbf{X}_t,C_{t-1},\mathbf{I}).
\tag{2}
$$

Thus, the position of the next ligand atom depends on its atom type. We regard both probabilities of the atom type and position as a joint

distribution over each preceding atom in $C_{t-1}$:

$$p(\mathbf{X}_t|C_{t-1},\mathbf{I}) \propto \prod_{i=1}^{t-1} p(\mathbf{X}_t|L_i,\mathbf{I}) \cdot \prod_{j} p(\mathbf{X}_t|P_j,\mathbf{I}), \tag{3}$$

$$p(\mathbf{r}_t|\mathbf{X}_t,C_{t-1},\mathbf{I}) \propto \prod_{i=1}^{t-1} p(d_{t,i}^{\text{ll}}|\mathbf{X}_t,L_i,\mathbf{I}) \cdot \prod_{j} p(d_{t,j}^{\text{lp}}|\mathbf{X}_t,P_j,\mathbf{I}), \tag{4}$$

where $d_{t,i}^{\text{ll}}$ and $d_{t,j}^{\text{lp}}$ are Euclidean distances between corresponding pairs of atoms, respectively. We assume that the type and position of a ligand atom mostly depend on its proximal protein atoms since a non-covalent interaction between a protein and a ligand is significant between closely contacting atom pairs. Hence, the probabilities conditioned on protein atoms can be approximated as follows:

$$\prod_{j} p(\mathbf{X}_t|P_j,\mathbf{I}) \simeq \prod_{j\in\mathcal{N}_k(t^*)} p(\mathbf{X}_t|P_j,\mathbf{I}), \tag{5}$$

$$\prod_{j} p(d_{t,j}^{\text{lp}}|\mathbf{X}_t,P_j,\mathbf{I}) \simeq \prod_{j\in\mathcal{N}_k(t^*)} p(d_{t,j}^{\text{lp}}|\mathbf{X}_t,P_j,\mathbf{I}), \tag{6}$$

where $\mathcal{N}_k(\cdot)$ yields the $k$-nearest neighboring pocket atom indices of a given ligand atom index. $t^*$ is an index of a ligand atom where the next atom will be added adjacent to, which is sampled from already placed ligand atoms at a time step $t$. We define $L_{t^*}$ as an atom-of-interest. This approximation enables the atom addition to be locally guided by surrounding pocket atoms and interaction conditions to enhance the possibility of constructing desirable protein–ligand interactions.

Meanwhile, the baseline model used to demonstrate the model generalizability is aimed at learning $p(\mathbf{L}|\mathbf{P})$ instead of $p(\mathbf{L}|\mathbf{P},\mathbf{I})$. All the other formulations are the same, except that the distributions are not dependent on $\mathbf{I}$ anymore.

## Model architecture of DeepICL

We adopt a variational auto-encoder (VAE) architecture[49] consisting of two main modules—an encoder and a decoder—as illustrated in Fig. 5. The encoder module embeds the structure of a given protein–ligand complex, $\mathbf{L}$ and $\mathbf{P}$, into a latent vector $\mathbf{z}$ that follows a standard normal distribution (see Fig. 5c). The decoder module then sequentially generates a ligand structure in an atom-wise manner from the latent vector $\mathbf{z}$ (see Fig. 5d). The interaction condition is integrated into the latent vector $\mathbf{z}$ for placing a suitable ligand atom to form desired interactions with the target. The encoder and decoder modules share the same embedding layers, which are composed of multiple layers of $E(3)$-invariant interaction network that propagates the messages between a protein and a ligand (see Fig. 5e). More details about the $E(3)$-invariant interaction network can be found in Supplementary section 1b.

DeepICL employs two additional dummy atoms that only hold positional information, the center-of-mass and the atom-of-interest, to assist the ligand design process as in the work of cG-SchNet[48]. The center-of-mass of the original ligand roughly determines the global position of a ligand to be generated. The atom-of-interest confines a 3D space where the next ligand atom would be placed; only its neighboring protein atoms are considered in predicting the next atom type and its position in each step. Consequently, DeepICL can learn the relationship between a local pocket environment and a structural preference of a ligand to fulfill the given interaction condition, leveraging the robustness of DeepICL in ligand design tasks for any protein. The above two dummy atoms are treated as individual ligand atoms in the training and sampling process. Then, they are removed when finalizing the ligand structure.

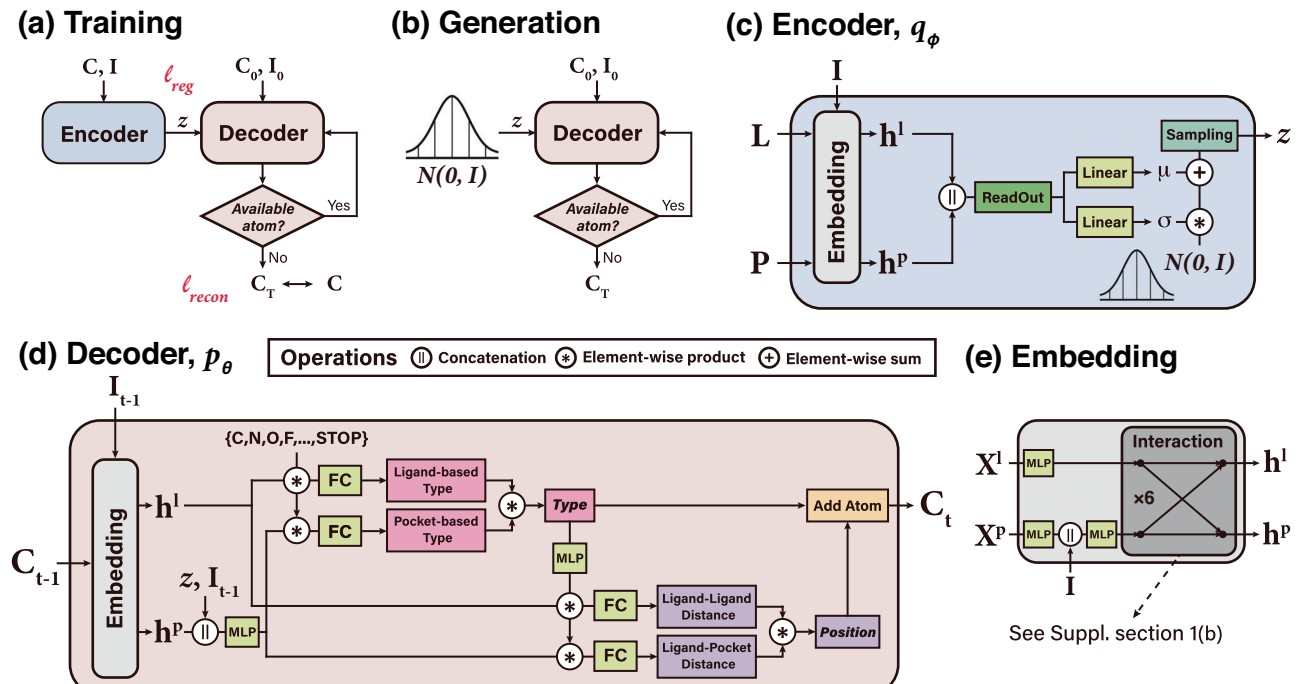

**Fig. 5 | Illustration of the model architecture of DeepICL. a** The training phase of DeepICL, where two losses $\ell_{\text{reg}}$ and $\ell_{\text{recon}}$ are denoted. **b** In the generation phase of DeepICL, $\mathbf{z}$ is sampled from the standard normal distribution instead of using the encoder. **c** The encoder module ($q_\phi$) is trained to encode a whole protein–ligand complex ($\mathbf{L}$, $\mathbf{P}$) and corresponding interaction condition, $\mathbf{I}$, into a latent vector $\mathbf{z}$ that follows a prior distribution. **d** The decoder module ($p_\theta$) is trained to reconstruct the ligand structure from the given protein pocket and an interaction condition with an autoregressive process. Note that the decoder of the figure describes a single atom addition step, where a type and a position of the $t$th ligand atom are determined from the protein–ligand complex of step $t-1$. **e** The embedding module is included in front of the encoder and decoder, incorporates interaction conditions to protein atoms, and updates protein and ligand atom features via interaction layers.

## Training DeepICL

The training objective of DeepICL is to predict the next atom, $L_t$, from a previous complex state, $C_{t-1}$, and the atom-of-interest. Since there is no canonical order in an atom-wise designing process, we randomly traverse a trajectory of placing atoms of a ligand in each training epoch. The next atom is always picked from the atoms covalently bonded to the atom-of-interest in the original ligand. DeepICL then learns the likelihood of a type of the next atom and its position.

For each step of an atom placement, DeepICL is trained to predict the next atom type, $\mathbf{X}_t$, and its position, $\mathbf{r}_t$, based on the previous complex state, $C_{t-1}$, the latent vector, $\mathbf{z}$, and interaction condition, $\mathbf{I}$. DeepICL embeds the information of $C_{t-1}$ into two sets of hidden vectors for the ligand and protein, $\mathbf{h}_{t-1}^l := \{\mathbf{h}_{t-1,i}^l\}$ and $\mathbf{h}_{t-1}^p := \{\mathbf{h}_{t-1,j}^p\}$, respectively. Again, $i$ and $j$ denote the atom indices of a ligand and a protein, respectively. We use two models for atom type prediction; one model, $\theta_l$, predicts the likelihood from already placed ligand atoms, and the other, $\theta_p$, predicts the likelihood from $k$-nearest neighboring protein pocket atoms. We minimize the Kullback-Leibler (KL) divergence between the predicted atom type distribution, $p_t^{\text{type}}$, and the ground-truth atom type distribution, $q_t^{\text{type}}$, which is a one-hot encoding of $\mathbf{X}_t$. Formally, we minimize the following type loss:

$$\ell_t^{\text{type}} = \underbrace{\frac{1}{t-1}\sum_{i=1}^{t-1} \text{KL}(p_{t,i}^{\text{type}}||q_t^{\text{type}})}_{\text{ligand-based type}} + \underbrace{\frac{1}{k}\sum_{j \in \mathcal{N}_k(t^*)} \text{KL}(p_{t,j}^{\text{type}}||q_t^{\text{type}})}_{\text{pocket-based type}}, \quad (7)$$

where $p_{t,i}^{\text{type}} = p_{\theta_l}(\mathbf{X}_t|\mathbf{h}_{t-1,i}^l,\mathbf{I},\mathbf{z})$, $p_{t,j}^{\text{type}} = p_{\theta_p}(\mathbf{X}_t|\mathbf{h}_{t-1,j}^p,\mathbf{I},\mathbf{z})$, and $N_k(t^*)$ denotes the number of $k$-nearest neighboring protein pocket atoms from the current atom-of-interest $t^*$. We also train the distance prediction model by minimizing the KL divergence loss for the distance distribution over the already placed ligand atoms and the proximal protein atoms:

$$\ell_t^{\text{dist}} = \underbrace{\frac{1}{t-1}\sum_{i=1}^{t-1} \text{KL}(p_{t,i}^{\text{dist}}||q_{t,i}^{\text{dist}})}_{\text{ligand-ligand distance}} + \underbrace{\frac{1}{k}\sum_{j \in \mathcal{N}_k(t^*)} \text{KL}(p_{t,j}^{\text{dist}}||q_{t,j}^{\text{dist}})}_{\text{ligand-pocket distance}}, \quad (8)$$

where $p_{t,i}^{\text{dist}} = p_{\theta_l}(d_{t,i}^{ll}|\mathbf{X}_t,\mathbf{h}_{t-1,i}^l,\mathbf{I},\mathbf{z})$ and $p_{t,j}^{\text{dist}} = p_{\theta_p}(d_{t,j}^{lp}|\mathbf{X}_t,\mathbf{h}_{t-1,j}^p,\mathbf{I},\mathbf{z})$. Here, $q^{\text{dist}}$ is a Gaussian expansion of a ground-truth distance, whose detailed definition can be found in Supplementary Equation (1).

We note that the training losses on pocket atoms incorporate only the $k$-nearest neighboring pocket atoms that are close to a ligand atom-of-interest so that the type of a newly added ligand atom is determined solely based on the surrounding local chemical environment. The atom type loss, $\ell_t^{\text{type}}$, and distance loss, $\ell_t^{\text{dist}}$, are minimized simultaneously to train the model to reconstruct a ligand structure. Thus, the reconstruction loss can be written as follows:

$$\ell_{\text{recon}} = \sum_t \left[ \ell_t^{\text{type}} + \ell_t^{\text{dist}} \right]. \quad (9)$$

The VAE architecture of DeepICL also requires a minimization of the following additional loss known as the regularization loss:

$$\ell_{\text{reg}} = \text{KL}(q_\phi(\mathbf{z}|\mathbf{L},\mathbf{P},\mathbf{I})||p(\mathbf{z})), \quad (10)$$

where $p(\mathbf{z})$ is the standard normal distribution.

## Designing ligands with DeepICL

DeepICL designs a ligand in three stages: (1) initialization, (2) sequential addition of atoms, and (3) termination of the process.

In the initialization stage, two additional dummy atoms, the center-of-mass and atom-of-interest, are combined into $C_0$ to guide the overall sampling process. The center-of-mass remains unmoved throughout the entire sampling process, whereas the atom-of-interest moves its position to one of the already placed ligand atoms in each addition step. Although one can manually select an arbitrary point as a starting point, in this work, we choose the center-of-mass of a reference ligand for convenience. To increase the diversity of generated ligands and decrease the dependency on the center-of-mass of the original ligand, we introduce a roto-translational Gaussian noise during a sampling phase. For the ligand elaboration task, where the generation starts from a pre-defined core structure, the initial structure is noised without the change in internal coordinates. More details about the Gaussian noise can be found in Supplementary section 3a.

In the second stage, DeepICL designs a ligand by sequentially adding new atoms. Based on the initialized state, DeepICL predicts the next atom type and its position in an autoregressive manner. Each likelihood of type and position comprises the likelihoods obtained from the ligand and protein sides, respectively. Thus, we integrate them as follows:

$$\log p(\mathbf{X}_t|C_{t-1},\mathbf{I}) \propto \sum_{i=1}^{t-1} \log p_{t,i}^{\text{type}} + \lambda \sum_{j \in \mathcal{N}_k(t^*)} \log p_{t,j}^{\text{type}}, \quad (11)$$

$$\log p(\mathbf{r}_t|\mathbf{X}_t,C_{t-1},\mathbf{I}) \propto \sum_{i=1}^{t-1} \log p_{t,i}^{\text{dist}} + \lambda \sum_{j \in \mathcal{N}_k(t^*)} \log p_{t,j}^{\text{dist}}. \quad (12)$$

Here, $\lambda$ is a pocket coefficient that tunes the contribution of a pocket in determining the next atom. The value of $\lambda$ is determined depending on how far an upcoming ligand atom is apart from pocket atoms. We tend to decrease the contribution of the pocket if the ligand atom is placed away from the pocket since the protein-ligand interaction occurs at a short range. Further details are included in Supplementary section 3b.

If DeepICL predicts the STOP sign for the next atom type, the current atom-of-interest $t^*$ is marked as unavailable and no longer selected as an atom-of-interest. Then, the next atom of interest, $(t+1)^*$, is sampled from a currently available set of ligand atoms and used for the next step. The sampling process terminates when every placed ligand atom is marked as unavailable, as illustrated in Fig. 5b. As a result, DeepICL yields a set of ligand atoms designed inside a target pocket. The bond orders are then inferred with OpenBabel software[50] to obtain a completed ligand structure.

## Interaction fingerprint and interaction similarity

We define interaction fingerprint and interaction similarity to evaluate how well the sampled ligands satisfy the given interaction condition. The interaction fingerprint describes the pattern of a protein's interaction with a specific ligand at an atom level. Each protein atom falls into one of four classes depending on the type of interaction it is involved in—hydrogen bond, hydrophobic interaction, salt bridge, and $\pi-\pi$ stacking. Unlike the interaction condition introduced in the Result section, the non-interaction class is neglected to build an interaction fingerprint. We then concatenate all the atom-wise one-hot vectors to obtain an interaction fingerprint as a single vector while preserving the atomic order in the protein. This ensures that the resulting interaction fingerprints can be compared across different ligands bound to a single target.

Next, we define interaction similarity as a cosine similarity between the interaction fingerprints of two ligands for a single target. To measure how well the ligand satisfies the given condition, we use the interaction fingerprint obtained from the original ligand as a reference to compare with those of the generated ligands. High interaction similarity indicates that the sampled

ligand possesses an interaction pattern similar to that of the original ligand. Hence, it follows the given interaction condition. With this interaction similarity metric, we can quantitatively evaluate the performance of our local interaction-aware conditioning strategy to control the ligand design process. We note that a low interaction similarity does not necessarily imply a low binding affinity for the sampled ligand. Still, the ligand may have the potential to form a better binding with a target by adopting a different interaction pattern than the original one.

### Ligand Elaboration

We focused on a ligand elaboration task in the Result sections. To obtain a core structure of a ligand, we removed chains and functional groups, leaving the key structure of the original ligand. We used the core ligand as an initial state of the generation stage. While demonstrating the effect of interaction conditioning, we elaborated the core ligands to generate 1000 ligands, respectively. The interaction conditions were established based on the original ligands. We then measured interaction similarities between the generated and original interaction patterns to select a ligand with the highest interaction similarity. In the binding pose stability analysis, the difference was that we used a reference-free interaction condition. This enables the formation of any possible interactions instead of relying on a specific condition extracted from the original ligand. After designing 1000 ligands for each pocket, we randomly sampled 10 ligands whose numbers of heavy atoms were the same as that of the original ligand to match the number of undergone generation steps.

### de novo ligand design

We performed a de novo ligand design task in the binding affinity analysis, the geometric analysis of generated interactions, and the analyses of chemical diversity and novelty. Here, a whole ligand structure is generated from a randomly initialized dummy atom instead of starting from a predefined ligand core structure. In those analyses, 100 whole ligands are designed from each of the test pockets. The center-of-masses of each original ligand in the reference complexes were used, with a Gaussian noise perturbation. We used reference-free interaction conditions during the Result sections for analyzing binding affinity and interaction geometries. In contrast, specific conditions from the references were used for investigating the chemical diversity and novelty to justify that our model can generate diverse and novel ligand structures even if the specific interaction pattern is constrained.

After the ligand generation, we locally optimized the structures and scored the binding affinities of those protein–ligand complexes with SMINA[42], a scoring and docking software based on AutoDock Vina[51], in the binding affinity analysis. Protein–ligand complexes in the training and test sets were also scored for later comparison. Note that we did not apply any further structure optimization in the geometric analysis of generated interactions since we were analyzing the generated intermolecular atom-atom distance distributions with that of the PDBbind 2020 ground-truth complexes.

### Mutant-selective epidermal growth factor receptor (EGFR) inhibitor design

We have demonstrated the design of mutant-selective inhibitors for EGFR with our generative framework, which is one of the most challenging problems in drug discovery. We retrieved complex structures of a wild-type and a double-mutated EGFR reported by Sogabe et al. (PDB ID: 3w2s and 3w2r, respectively)[52]. The two complexes share the same ligand and have similar pocket structures. To achieve selectivity toward the mutated EGFR, we assumed that a ligand that strongly interacts with the mutated

residues would favorably bind to the mutated pocket more than the wild-type. Under this assumption, we manually designated the possible interaction types of the atoms of MET790 and ARG858, the double-mutated residues, while sparing other atoms. Then, we underwent de novo ligand generation inside the double-mutated pocket (3w2r). After generating 1000 ligands inside the double-mutated EGFR, we also placed them in the aligned pocket of the wild-type EGFR. We then performed a local optimization followed by energy scoring via SMINA for each pocket.

## Data availability

The original protein–ligand complex data used in this study is available in the PDBbind database http://www.pdbbind.org.cn. The processed data for the generation are available at https://github.com/ACE-KAIST/DeepICL[53] Source data files relevant to each figure are provided with this paper. Source data are provided with this paper.

## Code availability

The implementation for our whole framework, including the training and sampling of DeepICL and evaluating generated ligands, is available at https://github.com/ACE-KAIST/DeepICL[53].

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

## Acknowledgements

This work was supported by Basic Science Research Programs through the National Research Foundation of Korea funded by the Ministry of Science and ICT (Grant No. 2018R1A5A1025208, 2023R1A2C2004376, and RS-2023-00257479) to W.Z., H.K. and W.Y.K.

## Author contributions

W.Z. conceptualized the work and developed the model. W.Z. trained the model and carried out the experiments. W.Z., H.K. and W.Y.K. designed the experiments, analyzed the results, and contributed to manuscript writing. The whole work was supervised by W.Y.K.

## Competing interests

The authors declare no competing interests.
