## [Peer Review File · Nature Communications]

REVIEWER COMMENTS

Reviewer #1 (Remarks to the Author):

The submitted paper describes a recurrent deep generative model for generating ligands conditional on a binding site (including a bound ligand) and optionally an initial ligand structure. It shows that including explicit interaction information to condition the model improves the quality of the generated ligands. The topic is interesting and data justifies the conclusions (the evaluations done show the interaction information is beneficial for the desired task). Although the manuscript is well written, my main concerns are with a lack of clarity/detail in some of the presentation that can likely be resolved with minor revisions.

I strongly recommend the authors re-work Fig 5, as the model is a key contribution of the paper, and the figure could better illustrate the model. For example, it would be helpful to know where GNN operations are applied versus MLPs. For instance, the manuscript (and supplement) are vague about how I is incorporated ("joined to z "). Looking at the code, it is combined with the hidden features of the pocket using an MLP to construct the input of the embedding layers and z , the hidden pocket features, and I are concatenated and reduced with an MLP after z is generated. These operations can't be deduced from the manuscript, figure, or supplement. Updating the figure to reflect the modularity and flow of the code (and clearly illustrate the recurrent nature of the model) would definitely strengthen the paper. You may need a subfigure for the architecture of your Embedder.

Fig 3. I am assuming the results in Fig B-E are for the full 100 protein test set and not just the 3 illustrative examples of 3a-b. If this is true, it should be made clear in the text and caption. Otherwise, the results should be updated to reflect the full test set - 3 examples isn't sufficient. The 3 overlaid histograms in Fig 3d-e are difficult to interpret - using kernel density estimates (lines) on one graph or 3 separate subplots would make the trends easier to see.

There needs to be a discussion/reporting of chemical validity. Looking at the code it seems that this was calculated and simply not reported. Is this why you do 1000 generation runs but only sample 10? Because some percentage fail? That's not a random sample.

The construction of the test and validation sets is well done, but I am confused by the condition that the number of protein heavy atoms is less than 300. This would leave only small peptides. Perhaps you mean pocket atoms? If so, how is a pocket determined?

It could be made clearer that the interaction information is an additional one-hot vector on each receptor atom.

Graph legends should be optimized to not cover up data.

In a couple places in the supplement you reference a section but the reference is missing.

Reviewer #2 (Remarks to the Author):

The authors propose a conditional generative model, DeepICL, to design ligands with desired interactions with a protein target. They train and evaluate their approach using targets from PDBbind.

I commend the goal of the authors and agree with the need for improved conditional generative models. I was encouraged by certain experimental choices (e.g. using protein and ligand similarity to separate train and test sets). However, I have significant reservations with other components of the study design and experimental conclusions.

My specific comments can be found below.

- The authors make several unsubstantiated claims. For example, in the abstract, they claim

“successful design of mutant-selective inhibitors” and in the introduction claim “we carried out real-world applications”. However, no experimental testing was performed.

- The manuscript was hard to follow in a number of places due to several Results sections lacking sufficient experiment details for the reader to understand. While I appreciate the methods section is provided at the end of the manuscript, the authors need to include the most critical details earlier in the manuscript.

- There is a lack of baseline approaches to which the authors proposed approach is compared. The authors selectively use 2 baselines that are ablations of the proposed method: (1) a masked interaction pattern and (2) a version of their model trained without interaction patterns. This does not allow us to rigorously validate the proposed model compared to other existing generative approaches or more straightforward baselines. In the case of “masked interaction conditions” were all conditions masked, or just some? The masking procedure was not well defined. Additionally, why did the author separately use masking and a version of their method trained without interaction patterns? It seemed strange that the authors used a new model for Fig. 3 b-e rather than using the masked conditions used for the results shown in Fig. 3a.

- Regarding the additional model, the authors state the model is “trained only with structural information, without any information about interaction types of protein atoms”. Please could the authors clarify what they mean? If the model is trained with structural information, then it has been trained with information about the protein atoms.

- Figure 1 - Stage 2 offers no insight into how the framework operates or uses the interaction patterns. Consequently, it does not aid the reader.

- P7 introduces the term “naive conditions”. Where are these specific conditions defined? Also, if these are the most detailed conditions used, why do the authors refer to them as “naive”?

- If I understand correctly, Section 4.1 states that the training ligands are from distinct protein targets. However, in Fig 3, the caption states “the training (black) and test (white) set for each pocket”. Please could the authors clarify? This is particularly important since the authors claim in several places the ability to “generalize”. However, the use of training ligands would not necessarily show generalization.

- Fig 3(c) - is the goal to reproduce the same number of interactions as the training set? Would a more interesting comparison not be with the interactions that were included in the conditions provided to the model? Additionally, these results are averaged across targets, which will have different desired numbers of interactions. Thus it is not apparent if closely matching the number of interactions per molecule on average meant that the number of interactions is indeed close to the desired number.

- (P10) “ For instance, if our model faces a situation where generating a hydrogen bond is favored, the model is likely to add nitrogen, oxygen, and fluorine instead of carbon despite carbon’s abundance in the training data, whereas the baseline model may also consider these atoms but with lower rates.” Please can you quantify these rates or provide some additional evidence for this claim?

- While I agree that chemical space is indeed large, it does not necessarily strike me as a positive that over 90% of the novel scaffolds were not present in the ChEMBL database (P11). This raises questions regarding the chemical validity and synthesizability of the generated molecules. How did the authors assess this? My concerns are supported by the generated molecule shown in Fig. 4b.

- Section 4.1 - did this procedure happen to leave exactly 100 test complexes or did the authors choose to sample this number?

- Section 4.2.2 - Please could the authors clarify the first stage? Determining suitable interaction conditions is a crucial step. I would encourage the authors to make their explanation when not using the ground truth more clear.

- Section 4.4 - "we removed chains and functional groups, leaving the key structure of the original ligand". What fraction of the molecule did this remove? Did this ever remove the entire molecule?

Finally, while not the basis of my review, I believe the manuscript would benefit from language editing. There are a number of sentences that do not make sense (e.g. "further manifests the impact"), are imprecise ("sample a bunch of protein-ligand complexes", "the chemical space is enormous"), or could be improved ("entire atoms").

REVIEWER COMMENTS

Reviewer #1 (Remarks to the Author):

The submitted paper describes a recurrent deep generative model for generating ligands conditional on a binding site (including a bound ligand) and optionally an initial ligand structure. It shows that including explicit interaction information to condition the model improves the quality of the generated ligands. The topic is interesting and data justifies the conclusions (the evaluations done show the interaction information is beneficial for the desired task). Although the manuscript is well written, my main concerns are with a lack of clarity/detail in some of the presentations that can likely be resolved with minor revisions.

1. I strongly recommend the authors re-work Fig 5, as the model is a key contribution of the paper, and the figure could better illustrate the model. For example, it would be helpful to know where GNN operations are applied versus MLPs. For instance, the manuscript (and supplement) are vague about how I is incorporated ("joined to z "). Looking at the code, it is combined with the hidden features of the pocket using an MLP to construct the input of the embedding layers and z , the hidden pocket features, and I are concatenated and reduced with an MLP after z is generated. These operations can't be deduced from the manuscript, figure, or supplement. Updating the figure to reflect the modularity and flow of the code (and clearly illustrate the recurrent nature of the model) would definitely strengthen the paper. You may need a subfigure for the architecture of your Embedder.

A: We appreciate the reviewer for pointing out the unclarity of the model illustration. We reworked Figure 5 and its caption, where each operation is depicted to describe the entire model structure. Moreover, how the interaction condition is incorporated into the encoder and decoder is specified. (Page 17)

Revised Figure 5. Illustration of the model architecture of DeepICL. (a) The training phase of Deep ICL, where two losses L_{reg} and L_{recon} are denoted. (b) In the generation phase of DeepICL, z is sampled from the standard normal distribution instead of using the encoder. (c) The encoder module (q_ϕ) is trained to encode a

whole protein-ligand complex (L, P) and corresponding interaction condition, I , into a latent vector z that follows a prior distribution. (d) The decoder module (p_{θ}) is trained to reconstruct the ligand structure from the given protein pocket and an interaction condition with an autoregressive process. Note that the decoder of the figure describes a single atom addition step, where a type and a position of the t -th ligand atom are determined from the protein-ligand complex of step $t-1$. (e) The embedding module is included in front of the encoder and decoder, incorporates interaction conditions to protein atoms, and updates protein and ligand atom features via interaction layers.

2. Fig 3. I am assuming the results in Fig B-E are for the full 100 protein test set and not just the 3 illustrative examples of 3a-b. If this is true, it should be made clear in the text and caption. Otherwise, the results should be updated to reflect the full test set - 3 examples isn't sufficient. The 3 overlaid histograms in Fig 3d-e are difficult to interpret - using kernel density estimates (lines) on one graph or 3 separate subplots would make the trends easier to see.

A: We appreciate the reviewer for the suggestion to improve the clarity. What the reviewer's first understanding is correct - three subplots in Fig. 3(a) are for the three representative examples, while (b)-(e) are for the full 100 test protein pockets. We added more information in both the text (page 10) and the caption. We also changed (d) and (e) to kernel density estimation (KDE) plots as suggested. (Page 9)

Changed captions are highlighted below:

Revised Figure 3. Demonstration of the generalizability of our generative framework. (a) Plots of ligand RMSDs during short MD simulations to assess the binding pose stability of designed ligands in three pockets from the test set - BMP1, FGF1, and DHFR. The blue and red curves depict the averaged RMSDs of ten sampled ligands of each generated set with 95% confidence intervals. Gray curves show ligand RMSDs of the original ligands. (b) The box plot shows the distribution of binding affinity scores. 100 ligands were generated with and without interaction information for each of the 100 test pockets, resulting in a total of 10,000 ligands. Their binding affinity scores were depicted in the blue and red boxes, respectively. The binding affinity scores of the ground-truth complexes composing the training and test sets are also analyzed, and depicted as the black and white boxes, respectively. Note that the training and test complexes are carefully separated, thus the training and test ligands are from distinct protein targets. The five-quartile values are displayed as boxes and whiskers. The average scores are shown as diamonds, while the bold lines in the middle of the boxes indicate the median scores of each set. (c) The bar plot shows the number of interactions per molecule for each interaction type from the generated complexes in (b). (d-e) Kernel density estimation plots of hydrophobic interaction and hydrogen bonding distances, respectively. The distances from the generated complexes in (b) were measured by using the PLIP software.

3. There needs to be a discussion/reporting of chemical validity. Looking at the code it seems that this was calculated and simply not reported. Is this why you do 1000 generation runs but only sample 10? Because some percentage fail? That's not a random sample.

A: Our model produces approximately 99% of chemically valid ligands, while the validity value may be affected by the randomness during the sampling process. We added the information in the supplementary information section 3. The updated sentence is as follows:

“Although the randomness of the sampling process affects the validity, our model produces approximately 99% of chemically valid ligands from the provided experimental settings in Supplementary Table 2.”

Moreover, since we obtained 9,930 valid ligand molecules by generating 10,000 ligands in section 2.4, we added the exact number to the manuscript, page 12.

In section 2.3, MD simulations for all the generated ligands are very demanding. Thus, we first filtered novel ligands to ensure that the numbers of their heavy atoms are the same as those of the reference ligand. We contemplated that the greater number of heavy atoms likely induces higher binding affinities. Thus, comparing ligands with the same atom numbers is necessary for fairness. This left less than 50 ligands for each target, and then we randomly sampled 10 ligands for the MD simulations for each target. We added more details about selecting samples for MD simulations on page 8, section 2.3.1. The updated sentences are as follows:

“We first filtered novel ligands to ensure that the numbers of their heavy atoms are the same as that of the reference ligand. We noted that the greater number of heavy atoms likely induces higher binding affinities. Thus, comparing ligands with the same atom numbers was necessary for fairness. This left less than 50 ligands for each target, and then we randomly sampled 10 ligands for MD simulations for each target.”

4. The construction of the test and validation sets is well done, but I am confused by the condition that the number of protein heavy atoms is less than 300. This would leave only small peptides. Perhaps you mean pocket atoms? If so, how is a pocket determined?

A: We apologize for the confusion due to our typo. We have corrected it to be 3,000 (about 160 amino acid residues) on page 14, section 4.1. This is because we wanted to exclude large proteins that need much longer time during the inferencing process or additional MD simulations.

5. It could be made clearer that the interaction information is an additional one-hot vector on each receptor atom.

A: We appreciate the reviewer for the suggestion and updated the context of defining the interaction information on page 3, section 2.1.1, and on page 8, section 2.3 for clearer information. The updated sentences are as follows:

“We define an interaction condition as a set of protein atoms’ additional interaction type one-hot vectors which indicates whether the atom can be involved in a particular interaction and its role in the interaction. (Page 3, section 2.1.1)”

“For the baseline comparison, we devised a model that was trained only on binding structures, without any explicit information about protein-ligand interactions, or to be more specific, without using additional interaction condition vectors for protein atoms. ... Here, we named sets of the generated ligands from the interaction-conditioned model and the baseline model with and without interaction information, respectively. (Page 8, section 2.3)”

6. Graph legends should be optimized to not cover up data.

A: We adjusted the legend sizes for all figures as suggested.

7. In a couple places in the supplement you reference a section but the reference is missing.

A: We thank the reviewer for the detailed review. We have revised the supplementary information as suggested.

Reviewer #2 (Remarks to the Author):

The authors propose a conditional generative model, DeepICL, to design ligands with desired interactions with a protein target. They train and evaluate their approach using targets from PDBbind.

I commend the goal of the authors and agree with the need for improved conditional generative models. I was encouraged by certain experimental choices (e.g. using protein and ligand similarity to separate train and test sets). However, I have significant reservations with other components of the study design and experimental conclusions.

My specific comments can be found below.

1. The authors make several unsubstantiated claims. For example, in the abstract, they claim “successful design of mutant-selective inhibitors” and in the introduction claim “we carried out real-world applications”. However, no experimental testing was performed.

A: We appreciate the reviewer for pointing out the possible misunderstanding claims. Since no experimental testing was performed in our scope, we revised the expressions to reflect what we have actually done. The updated sentences are as follows:

“Moreover, the effective design of potential mutant-selective inhibitors demonstrates the unique applicability of our approach to structure-based drug design. (Page 1, abstract)”

“Finally, we applied our model to tackle practical problems where specific interaction sites play a crucial role, demonstrating the unique applicability of our approach to structure-based drug design. (Page 3, introduction)”

2. The manuscript was hard to follow in a number of places due to several Results sections lacking sufficient experiment details for the reader to understand. While I appreciate the methods section provided at the end of the manuscript, the authors need to include the most critical details earlier in the manuscript.

A: We admit that some critical details are only mentioned in the method section at the end of the manuscript. We thoroughly revised the manuscript to include the major experimental details in the result and discussion section. (Pages 3, 4, 6, 8)

3. There is a lack of baseline approaches to which the authors proposed approach is compared. The authors selectively use 2 baselines that are ablations of the proposed method: (1) a masked interaction pattern and (2) a version of their model trained without interaction patterns. This does not allow us to rigorously validate the proposed model compared to other existing generative approaches or more straightforward baselines.

A: To our best knowledge, the previous works for DL-based SBDD employed the CrossDocked2020 dataset containing 22.5 million computationally prepared

protein-ligand pairs, indicating that the reliability of the previous methods is limited to the accuracy of docking software rather than the amount of data. However, we trained our model with a much smaller database containing only about 19,000 experimentally validated structures, because the experimental data can be regarded as the most reliable ground truth, while computer-generated data has a critical issue in accuracy. Thus, we believe that the previous models are inappropriate as a baseline since the main purpose of this work is to improve the generalizability of the generative model trained with ground-truth small data.

Alternatively, in the revised manuscript, we devised a new baseline model where the model architecture and protein-ligand pairs in the training set are the same as our original model, but interaction conditions were not informed during both the training and generation phases. (see Section 2.3 on page 8) This provides a precise ablation of an interaction-aware conditioning strategy in ligand-designing tasks, having proved that the mere structural information is insufficient to design ligands with desirable properties, especially for unseen targets. Thus, we believe our baseline approach is effective in demonstrating the contribution of our strategy to improving the generalizable SBDD in a circumstance where an appropriate external baseline is unavailable, providing empirical evidence of benefits brought by incorporating the proper interaction information.

In the case of “masked interaction conditions” were all conditions masked, or just some? The masking procedure was not well defined.

A: All conditions were masked, so that a zero vector was fed in the position of the interaction condition vector as a “masked interaction condition”. We revised the main text for a more specific definition of the term “masked interaction conditions” as below.

“We fed our model a zero vector with the same size as the original interaction condition as a masked interaction condition. (Page 6, section 2.2)”

Additionally, why did the author separately use masking and a version of their method trained without interaction patterns? It seemed strange that the authors used a new model for Fig. 3 b-e rather than using the masked conditions used for the results shown in Fig. 3a.

A: We admit that it is not reasonable to use different baselines for binding stability analysis (Fig. 3(a)). A model trained without using a conditioning scheme was a key baseline to validate the effect of explicit incorporation of protein-ligand interactions. Thus, we reworked section 2.3.1 by using the model without using interaction information as a baseline instead of using masked interaction conditions, thus unifying the baseline model in Fig. 3 (a) to (e). (Page 8, 9, section 2.3.1) Please see also the reply to comment 3.

- Regarding the additional model, the authors state the model is “trained only with structural information, without any information about interaction types of protein atoms”. Please could the authors clarify what they mean? If the model is trained with structural information, then it has been trained with information about the protein atoms.

A: We appreciate pointing out the unclarity. Training only on protein-ligand complex structures would result in the baseline model that solely reproduces the geometries of binding structures. This baseline model might learn some information related to non-covalent interactions based on the atom occurrences but is less likely to be generalized on the typical patterns of those interactions due to the limited number of experimental protein-ligand pairs. Therefore, the baseline model inevitably relies on the statistical distribution of protein-ligand binding geometries in determining the type and position of a newly added atom. As a strategy to improve generalizability, we informed interaction types as a condition in addition to the structural information, which is prior knowledge of the protein-ligand interaction. We updated the corresponding context in detail on page 8, head of section 2.3.

The updated paragraph is as follows:

“This baseline model might learn some information related to non-covalent interactions based on the atom occurrences but is less likely to be generalized on the typical patterns of non-covalent interactions due to the limited number of protein-ligand pairs in the training set. Therefore, the baseline model inevitably relies on the statistical distribution of protein-ligand binding geometries in determining the type and position of a newly added atom.”

- Figure 1 - Stage 2 offers no insight into how the framework operates or uses the interaction patterns. Consequently, it does not aid the reader.

A: We appreciate the reviewer for the suggestion, we revised Figure 1 in detail to aid the better understanding of how the two main stages of our framework operate. (Page 5)

(a) Stage 1: Interaction-aware condition setting

(b) Stage 2: Interaction-aware 3D molecular generation

Revised Figure 1. A conceptualized illustration of our proposed interaction-aware 3D ligand generative framework. (a) The first stage profiles a protein pocket to designate an interaction condition on each protein atom. (b) In the second stage, DeepICL sequentially adds ligand atoms inside a protein pocket based on the predetermined interaction conditions. Letters inside circles indicate interaction types as follows: hydrogen bonds (**H**), hydrophobic interactions (**D**), salt bridges (**S**), and π - π stackings (π).

6. P7 introduces the term “naive conditions”. Where are these specific conditions defined? Also, if these are the most detailed conditions used, why do the authors refer to them as “naive”?

A: We admit that the term “naive conditions” was vague. Also, related to comment 3, we admit that the baseline model in binding stability analysis should be a model trained without an interaction conditioning scheme. Thus, we reworked our experiment with a new baseline model and thus changed the terms “naive” and “masked” condition into the model “with” and “without” the interaction information. (Page 8)

7. If I understand correctly, Section 4.1 states that the training ligands are from distinct protein targets. However, in Fig 3, the caption states “the training (black) and test (white) set for each pocket”. Please could the authors clarify? This is particularly important since the authors claim in several places the ability to “generalize”. However, the use of training ligands would not necessarily show generalization.

A: We apologize for the confused expression. Indeed, the training and test ligands are from distinct protein targets. We revised the caption of Figure 3 on page 9 as follows:

“The binding affinity scores of the ground-truth complexes composing the training and test sets are also analyzed, and depicted as the black and white boxes, respectively. Note that the training and test complexes are carefully separated, thus the training and test ligands are from distinct protein targets.”

8. Fig 3(c) - is the goal to reproduce the same number of interactions as the training set? Would a more interesting comparison not be with the interactions that were included in the conditions provided to the model? Additionally, these results are averaged across targets, which will have different desired numbers of interactions. Thus it is not apparent if closely matching the number of interactions per molecule on average means that the number of interactions is indeed close to the desired number.

A: We appreciate the reviewer for pointing out our ambiguous claim in the discussion part. Throughout the manuscript, we define the generalization ability as the capability of designing a potent ligand for unseen targets by forming favorable interaction patterns learned from the training phase, leading to achieving high binding stabilities and affinities. Fig. 3(b) shows our model could achieve the main goal of providing potent ligands with comparable binding affinities to those of the reference ligands

(experimental data), while Fig. 3(c) shows how the model successfully designed them by forming desired non-covalent interactions.

The desired numbers of interactions are different across the targets as the reviewer pointed out. Moreover, since the targets used in training and generation were distinctly split, comparing the number of interactions between those ligands would not be appropriate. Thus, we excluded the comparison with the training set in Fig. 3(c) on Page 10. Thanks to the reviewer's comment 9, deeper analysis of the hydrogen bonds via investigation of the frequencies of generated ligand atom types near the pockets' hydrogen donor or acceptor-typed atoms provides more direct evidence of increased hydrogen bond counts.

9. (P10) “ For instance, if our model faces a situation where generating a hydrogen bond is favored, the model is likely to add nitrogen, oxygen, and fluorine instead of carbon despite carbon’s abundance in the training data, whereas the baseline model may also consider these atoms but with lower rates.” Please can you quantify these rates or provide some additional evidence for this claim?

A: As the reviewer suggested, we quantified the generation rates of each atom type involved in the hydrogen bonds and provided them in Supplementary Table 4. We could strengthen our claim by showing that the usage of interaction information assisted the model in selecting more nitrogen, oxygen, and fluorine with a higher ratio in a situation to form hydrogen bonds. (Page 11)

Frequencies (%)	Carbon	Nitrogen	Oxygen	Fluorine
With information	63.3	14.5	21.8	0.0418
Without information	72.7	6.30	19.8	0.0205

Supplementary Table 4. The frequencies of ligand atom types that were generated within 4Å of hydrogen bond donor or acceptor protein atoms.

10. While I agree that chemical space is indeed large, it does not necessarily strike me as a positive that over 90% of the novel scaffolds were not present in the ChEMBL database (P11). This raises questions regarding the chemical validity and synthesizability of the generated molecules. How did the authors assess this? My concerns are supported by the generated molecule shown in Fig. 4b.

A: We appreciate the reviewer's comment. We have newly evaluated the synthesizability of the generated molecules by adopting SAscore, a widely used metric for synthetic accessibility introduced by Ertl and Schuffenhauer [1]. The calculated SAscores exhibited a very similar distribution to that of bioactive molecules introduced in the work of Ertl; the average SAscore of the generated molecules was 3.18, which is close to the peak of the graph for bioactive molecules (see Supplementary Fig. 5). This result suggests that in terms of SAscore, the synthetic complexity of the generated molecules is comparable to that of typical bioactive molecules. Still, this result does not fully guarantee the actual synthesizability of the molecules generated by the DeepICL model. However, the SAscore can effectively capture structural complexity and the presence of rarely appearing substructures.

Therefore, from the distribution of the SAScore, we conclude that most molecules designed by DeepICL have relatively similar structural complexities to those of typical bioactive molecules. We added this discussion with the SAScore distribution to the manuscript on page 12, section 2.4, and Supplementary Section 10 with Supplementary Figure 5.

Distribution of SAScore for the valid molecules generated by DeepICL and the molecules of the training data from PDBbind.

Distribution of SAScore for natural products, bioactive molecules and molecules from catalogues.[1]

For reference, the SMILES representation of that molecule shown in Fig. 4(b) is “NCC1=C2C3=C(N=C(/C(C=CC4)=C/CC(C(NC(O)=O)=O)C)C4=C3C(O)CC2)C=C1”. Its scaffold is a fused-ring system involving four six-membered rings of two aromatic but two aliphatic ones. We understand the reviewer’s concern about the synthesizability of this molecule, but the reason we chose this sample is that it clearly exhibited strong interactions at appropriate distances with targeted residues. The design of more synthetically accessible ligands was not a key concern in this example study because we focus on selectivity. But we agree that in real applications both selectivity and synthetic accessibility are important.

[1] Ertl, Peter, and Ansgar Schuffenhauer. "Estimation of synthetic accessibility score of drug-like molecules based on molecular complexity and fragment contributions." *Journal of cheminformatics* 1 (2009): 1-11.

11. Section 4.1 - did this procedure happen to leave exactly 100 test complexes or did the authors choose to sample this number?

A: We obtained exactly 109 test protein-ligand complexes from the data filtering and splitting. For convenience, we randomly dropped 9 complexes and only used 100 test complexes. We additionally provide 100 PDB IDs of test complexes in the supplementary information. We also added descriptions in the manuscript on page 14 as follows:

“We filtered out the rest of the data to leave 109 test complexes that satisfy the following three conditions ... For convenience, we randomly chose 100 test complexes from them, where their PDB IDs are provided in Supplementary section 12.”

12. Section 4.2.2 - Please could the authors clarify the first stage? Determining suitable interaction conditions is a crucial step. I would encourage the authors to make their explanation when not using the ground truth more clear.

A: We thank the reviewer for pointing out the unclarity of the first stage - an interaction-aware condition setting process. We leveraged two strategies for determining interaction conditions, depending on whether the information of binding ligands is available to use as a reference. If the reference does not exist, we analyzed the pocket residue atoms to determine the suitable interaction type in a rule-based fashion. On the other hand, we used the PLIP software to extract the reference condition from the reference protein-ligand binding structure. We moved the corresponding context to section 2.1.1 on Page 3~4 and updated the explanation in detail. Also, we revised Figure 1 for a better understanding of the first stage.

13. Section 4.4 - “we removed chains and functional groups, leaving the key structure of the original ligand”. What fraction of the molecule did this remove? Did this ever remove the entire molecule?

A: During the ligand elaboration, we need to pre-set the core structure. One option is to use the Bemis-Murcko scaffold, however, the scaffold often includes critical functional groups that are already forming non-covalent interactions with target proteins. In order to demonstrate the ability of our model to elaborate the core structures with almost no meaningful interactions, we manually removed functional groups, leaving the minimum cores composed of single or double rings. The corresponding contexts are as follows:

“Core structures were determined based on our visual inspection, removing chains and functional groups to leave the minimal structures composed of single or double rings as shown in Figure 2(a) and Supplementary Figure 1. (Page 5, section 2.2)”

14. Finally, while not the basis of my review, I believe the manuscript would benefit from language editing. There are a number of sentences that do not make sense (e.g. “further manifests the impact”), are imprecise (“sample a bunch of protein-ligand complexes”, “the chemical space is enormous”), or could be improved (“entire atoms”).

A: We carefully proofread the entire manuscript to improve the writing.

REVIEWERS' COMMENTS

Reviewer #1 (Remarks to the Author):

The authors were response to reviewer feedback and the result is a much improved paper.

Reviewer #1 (Remarks on code availability):

I looked through the github and it seems complete, but did not actually try to use the code.

Reviewer #2 (Remarks to the Author):

I thank the authors for their revisions. I believe they have improved the clarity and generally strengthened the manuscript.

Regarding baselines, while I would have preferred to see external baselines, I am satisfied that the experiments performed support the key claim of the manuscript, which is the benefit of the interaction information.

I have no further comments.

Response to Review 1.

We thank the reviewer for the careful evaluation and valuable comments that have helped us improve the quality of our manuscript.

Response to Review 2.

We thank the reviewer for the careful evaluation and valuable comments that have helped us improve the clarity and generally strengthen the manuscript. The experiments suggested by the reviewer backed the key claim of the manuscript by demonstrating the benefit of using interaction information in DL-based drug design.